# ST²360D: Spatial-to-Temporal Consistency for Training-free 360 Monocular Depth Estimation

Zidong Cao[1*]    Jinjing Zhu[1*]    Hao Ai[2*]    Lutao Jiang[1]    Yuanhuiyi Lyu[1]    Hui Xiong[1,3†]

[1]Thrust of Artificial Intelligence, HKUST (Guangzhou), China
[2]University of Birmingham, UK
[3]Department of Computer Science and Engineering, HKUST, Hong Kong SAR, China

caozidong1996@gmail.com; jinjingzhu.mail@gmail.com;
aihao199712@gmail.com; jianglutao98@gmail.com;
ryan.lyu.mail@gmail.com; xionghui@ust.hk

## Abstract

$360°$ monocular depth estimation plays a crucial role in scene understanding owing to its $180° \times 360°$ field-of-view (FoV). To mitigate the distortions brought by equirectangular projection, existing methods typically divide $360°$ images into distortion-less perspective patches. However, since these patches are processed independently, depth inconsistencies are often introduced due to scale drift among patches. Recently, video depth estimation (VDE) models have leveraged temporal consistency for stable depth predictions across frames. Inspired by this, we propose to represent a $360°$ image as a sequence of perspective frames, mimicking the viewpoint adjustments users make when exploring a $360°$ scenario in virtual reality. Thus, the spatial consistency among perspective depth patches can be enhanced by exploiting the temporal consistency inherent in VDE models. To this end, we introduce a *training-free* pipeline for $360°$ monocular depth estimation, called **ST²360D**. Specifically, ST²360D transforms a $360°$ image into perspective video frames, predicts video depth maps using VDE models, and seamlessly merges these predictions into a complete $360°$ depth map. To generate sequenced perspective frames that align with VDE models, we propose two tailored strategies. First, a spherical-uniform sampling (**SUS**) strategy is proposed to facilitate uniform sampling of perspective views across the sphere, avoiding oversampling in polar regions typically with limited structural details. Second, a latitude-guided scanning (**LGS**) strategy is introduced to organize the frames into a coherent sequence, starting from the equator, prioritizing low-latitude slices, and progressively moving toward higher latitudes. Extensive experiments demonstrate that ST²360D achieves strong zero-shot capability on several datasets, supporting resolutions up to 4K.

## 1   Introduction

There has been growing interest in $360°$ cameras for capturing a $180° \times 360°$ field-of-view (FoV). $360°$ monocular depth estimation is crucial for scene understanding [1, 2] with a wide range of applications such as autonomous driving [3, 4], virtual reality (VR) [5, 6, 7], and visual navigation [8]. The most common representation of $360°$ images is the equirectangular projection (ERP), which maps spherical data onto a 2D plane for storage and processing. However, ERP representation introduces spherical distortions, particularly near the poles [9], stretching visual content unevenly and degrading the performance of perspective depth estimators when applied to $360°$ images.

---

*Equal contributions.
†Corresponding Author.

39th Conference on Neural Information Processing Systems (NeurIPS 2025).

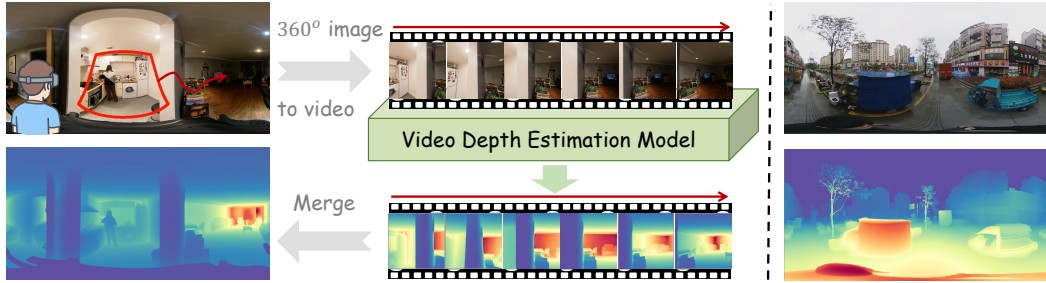

Figure 1: **Left:** Our proposed training-free pipeline. We represent a 360° image as video frames, mimicking viewpoint adjustments in VR. The arrow $\rightarrow$ denotes the direction of the visual transition. The video frames are processed with the VDE model to predict video depth maps, which are merged into a 360° depth map. **Right:** ST²360D shows impressive zero-shot capability in outdoor scenario.

To mitigate distortions, existing methods mainly project 360° images onto distortion-less perspective patches [10, 11, 12, 13, 14], which are more compatible with standard perspective models. However, processing these perspective patches independently often leads to depth inconsistencies, causing noticeable seams and degraded performance. To address this issue, both training-based and training-free methods have been proposed. Training-based methods [10, 14] fuse global features from ERP images with local features extracted from perspective patches. However, these methods heavily rely on expensive 360° depth annotations [15, 16, 17], and have limited zero-shot capabilities. Furthermore, due to limited computational resources, the input resolution for ERP images is constrained. In contrast, training-free methods [18, 19] convert high-resolution ERP images into perspective patches, and estimate patch-wise depths in parallel using pre-trained perspective models [20, 21]. Subsequently, the perspective depth patches are re-projected to the ERP plane and merged into a high-resolution 360° depth map. Nevertheless, these methods still face challenges, such as dependence on time-consuming post-processing [18] or requiring additional 360° depth estimation models as references [19].

Recently, Video Depth Estimation (VDE) models [22, 23, 24] have demonstrated significant advancements, enabling stable and accurate depth predictions across video frames. Specifically, VDE models ensure depth alignment within overlapping regions of consecutive frames by leveraging temporal consistency, significantly mitigating depth flickering issues. This raises an interesting question: *Can the temporal consistency inherent in VDE models be leveraged to address the depth inconsistency challenges encountered by current* 360° *methods?* To explore this, we propose representing a single 360° image as a sequence of perspective frames, closely mimicking the continuous viewpoint adjustments that occur in VR (See left of Fig. 1). Within VR environments, users experience smooth visual transitions as their viewpoints gradually shift, producing coherent sequences of perspective images akin to a continuous video stream. With this video representation, the spatial consistency among perspective depth patches can be enhanced by the temporal consistency in VDE models. To this end, we propose a novel training-free pipeline for 360° monocular depth estimation, named **ST²360D**. Specifically, ST²360D represents a 360° image as a sequence of perspective frames, then predicts video depth maps using VDE models, and finally seamlessly merges video depth maps back into the ERP plane to yield a complete 360° depth map.

To better align sequenced perspective frames with VDE models in our pipeline, we introduce two key strategies. First, we propose a spherical-uniform sampling (**SUS**) strategy, distributing viewpoints evenly across the sphere based on the vertices of a subdivided icosahedron. Compared with conventional uniform sampling on the ERP plane, our strategy avoids oversampling polar regions with limited structural information. Second, we introduce a latitude-guided scanning (**LGS**) strategy that organizes the viewpoints to obtain perspective frames. Given that VDE models rely on initial frames for accurate subsequent predictions [22], we begin scanning at equatorial viewpoints with rich structural information and sequentially search subsequent viewpoints from their spherical neighborhoods. In addition, we partition the spherical surface into latitude slices. The scanning path originates from the lowest-latitude slice; after fully traversing viewpoints within one slice, the scanning path progressively shifts upward to higher latitudes. This ensures a smooth, coherent, and structurally informative sequence, enhancing the performance of VDE models (See right of Fig. 1). Experiments demonstrate that ST²360D predicts consistent 360° depth maps with impressive zero-shot performance. Project page: https://caozidong.github.io/ST2360D_Depth/.

In summary, our contributions are three-fold: (I) We propose to represent a 360° image as a sequence of perspective frames, unleashing the temporal consistency of VDE models to enhance spatial consistency across perspective depth patches. (II) We present $ST^2360D$, a training-free pipeline for 360° monocular depth estimation, along with two key strategies—SUS and LGS—to better align with VDE models. (III) Extensive experiments demonstrate that $ST^2360D$ achieves impressive zero-shot performance across diverse scenarios, supporting resolutions up to 4K.

## 2 Related Work

### 2.1 360 Monocular Depth Estimation

To mitigate spherical distortions, existing 360° monocular depth estimation approaches typically divide 360° images into perspective patches, categorized into training-based and training-free methods.

**Training-based methods.** With publicly available 360° depth datasets [15, 16, 17], training-based methods have been developed and achieved promising performance. Besides ERP-based methods [25, 26, 27, 28, 29, 30], several methods adopt distortion-less perspective patches as input, including cubemaps (CP) [10, 11, 12] and tangent patches (TP) [13, 14]. However, these perspective patches sacrifice the global continuity in ERP, often leading to ambiguous depth scales and shifts among different patches. To address this, OmniFusion [13] introduces a geometry-aware fusion mechanism that integrates 3D geometric cues with patch features, while HRDFuse [14] collaboratively learns holistic contextual features from ERP images and regional structural details from TPs. Nevertheless, these methods heavily rely on labeled 360° depth datasets, which have a limited number of samples and are primarily composed of indoor scenes. Consequently, these training-based methods have limited zero-shot capabilities, particularly for outdoor scenarios. To address it, recent approaches, including PanDA [25] and Depth Anywhere [31], have leveraged the perspective depth foundation models [32, 33] to generate pseudo depth labels for unlabeled 360° images. However, taking ERP images as input constrains them for high-resolution 360° depth estimation due to limited GPU memory. *Instead, our $ST^2360D$ is training-free and can be flexibly extended to 4K resolutions.*

**Training-free methods.** Recently, several training-free pipelines [18, 19] have been proposed. The general pipeline is to project a single 360° image into multiple perspective patches, leverage pre-trained models [20] to predict perspective depth patches in parallel, and subsequently re-project these patches onto the ERP plane to form a complete 360° depth map. 360MonoDepth [18] proposes a deformable multi-scale alignment to recombine the individual depth patches. However, such post-processing is computationally expensive, particularly at high resolutions. A subsequent method [19] introduces a pre-trained 360° depth estimator to generate an initial ERP depth map as reference, which is utilized for perspective depth patches to register to. Nevertheless, this referenced estimator still requires annotated 360° depth data for supervised training. *In contrast, our $ST^2360D$ addresses the depth inconsistencies among patches by exploiting the temporal consistency in VDE models.*

### 2.2 Video Depth Estimation

Unlike image-based depth estimation methods [32, 33, 34, 35, 36], VDE methods aim to maintain temporal consistency, specifically by minimizing flickering effects between consecutive frames. Current VDE methods fall into two main categories: test-time optimization and feed-forward prediction. Test-time optimization approaches [37, 23, 38] fine-tune a pre-trained image-based depth estimation model on testing videos, typically requiring auxiliary information like camera poses or optical flow during inference. And feed-forward prediction methods [39, 40, 41] are trained directly on video datasets, leveraging both spatial and temporal supervision. Recent advancements, including ChronoDepth [42], DepthCrafter [43], and DepthAnyVideo [24], employ pre-trained video diffusion models (*e.g.*, Stable Video Diffusion [44]) to improve both stability and accuracy in video depth predictions. Furthermore, methods such as Video Depth Anything (VDA) [22] and BufferAnytime [45] extend the capabilities of the vision foundation models [32] while ensuring temporal stability over long video sequences. *To the best of our knowledge, $ST^2360D$ is the first to enhance the spatial consistency across perspective patches by leveraging the temporal consistency in VDE models.*

## 3 Methodology

Our $ST^2360D$ is a training-free pipeline for effective 360° depth estimation, leveraging the inherent temporal consistency in VDE models. We first briefly introduce the concepts of perspective patch

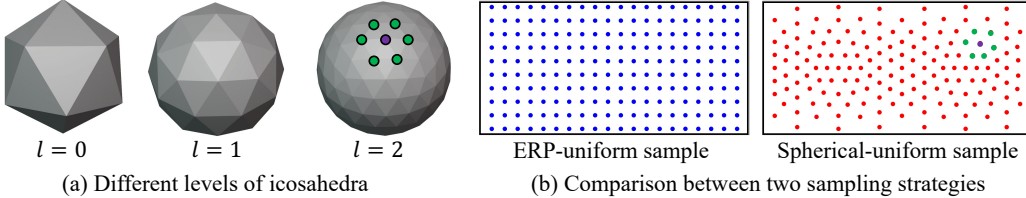

| $l = 0$ | $l = 1$ | $l = 2$ | ERP-uniform sample | Spherical-uniform sample |

(a) Different levels of icosahedra  (b) Comparison between two sampling strategies

Figure 2: **(a)** Different levels of icosahedra, whose vertices are utilized for sampling viewpoints. **(b)** Compared with ERP plane sampling, the proposed spherical-uniform sampling strategy prevents oversampling near poles. • vertices are the spherical neighbors of the • vertice.

projection in Sec. 3.1. To compile these perspective patches into video frames that align with VDE models, we propose two tailored strategies: spherical-uniform sampling (SUS) in Sec. 3.2 and latitude-guided scanning (LGS) in Sec. 3.3. The SUS strategy distributes the perspective patches evenly on the spherical surface, preventing oversampling near the poles with limited structural details. Moreover, the LGS strategy prioritizes perspective patches in the low-latitude slices and gradually progresses toward high-latitude slices. Finally, in Sec. 3.4, we utilize VDE models to predict video depth maps from the video frames and merge the video depth maps into a complete $360°$ depth map.

## 3.1 Preliminary of Perspective Patch Projection

Given a $360°$ image in ERP format $\mathbf{I} \in \mathbb{R}^{H \times W \times 3}$, we extract $N$ perspective patches $\mathbf{P} = \{\mathbf{P}_1, \ldots, \mathbf{P}_N\}$ based on $N$ viewpoints $\mathbf{v} = \{\mathbf{v}_1, \ldots, \mathbf{v}_N\}$ on the spherical surface, each with spatial resolution $S \times S$. The perspective projection from $\mathbf{I}$ to $\mathbf{P}$ involves three primary steps. Firstly, we determine parameters for $N$ virtual cameras, including intrinsic and extrinsic matrices. The intrinsic matrix $\mathbf{K}$ is defined by the focal length $f$ which is derived from the FoV $\alpha$: $f = \frac{1}{2\tan(\alpha/2)}$, with the center point set to $c_x = c_y = \frac{S-1}{2}$. Note that all virtual cameras share this intrinsic matrix. For the extrinsic matrix of the $i$-th virtual camera, denoted $\mathbf{E}_i = [\mathbf{R}_i, \mathbf{t}_i]$, the rotation matrix $\mathbf{R}_i$ transforms coordinates from the world coordinate system to camera coordinate system, determined primarily by the viewpoint $\mathbf{v}_i$. The translation matrix $\mathbf{t}_i$ is zero. Secondly, we construct a pixel coordinate map $\mathbf{X} \in \mathbb{R}^{H \times W \times 2}$ containing normalized image coordinates $(u, v) \in (0, 1)$. We project $\mathbf{X}$ from the image coordinates to the world coordinate system based on $\mathbf{K}$. The world coordinate system is then transformed to the virtual camera coordinate system based on $\mathbf{E}_i$: $[x, y, z]^T = \mathbf{E}_i \mathbf{K}[u, v, 1]^T$. Finally, these virtual camera coordinates are projected onto spherical coordinates $(\theta, \phi)$ using the following formula: $[\theta, \phi]^T = [\arcsin(z), \arctan2(y/x)]^T$. Using the calculated spherical coordinates as indices, pixels from the original ERP image $\mathbf{I}$ are sampled to generate the perspective patches. The entire process for generating patches $\mathbf{P}$ can be formulated as: $\mathbf{P}_i = \mathcal{P}(\mathbf{I}, \mathbf{v}_i, S, \alpha), i = 1, 2, ..., N$.

## 3.2 Spherical-Uniform Sampling (SUS) Strategy

To obtain viewpoints $\mathbf{v}$ for perspective patch projection, a straightforward method is to uniformly sample viewpoints on the ERP plane, as illustrated in Fig. 2(b). These sampled viewpoints are arranged from top to bottom to form a video sequence. However, this planar sampling strategy leads to suboptimal results (See Fig. 6), primarily due to redundant sampling in polar regions, which typically contain limited structural details (*e.g.*, ceilings, floors, or sky), as depicted in Fig. 3. Furthermore, since VDE models process fixed-length video frame segments, the initial segments may be dominated by less informative polar regions. As a result, although denser sampling potentially enhances video stability, excessive sampling near the poles can degrade the overall performance.

To address this, we propose a novel spherical-uniform sampling (SUS) strategy by leveraging icosahedron projection (ICOSAP) to evenly distribute viewpoints across the sphere (See Fig. 2(a)). An icosahedron approximates a spherical surface with significantly reduced distortion, especially as the subdivision level increases [46, 47]. At subdivision level $l$, the underlying icosahedral grid comprises $20 \times 4^l$ triangular faces and $10 \times 4^l + 2$ vertices. Practically, we utilize these vertices as candidate viewpoints for generating perspective patches. To prevent redundant sampling in regions of limited structural detail, vertices located excessively close to the poles (with absolute latitude exceeding $89°$) are excluded. The SUS strategy provides two specific benefits in our pipeline: (1) it reduces redundant sampling in polar regions, reallocating viewpoints towards low-latitude regions that

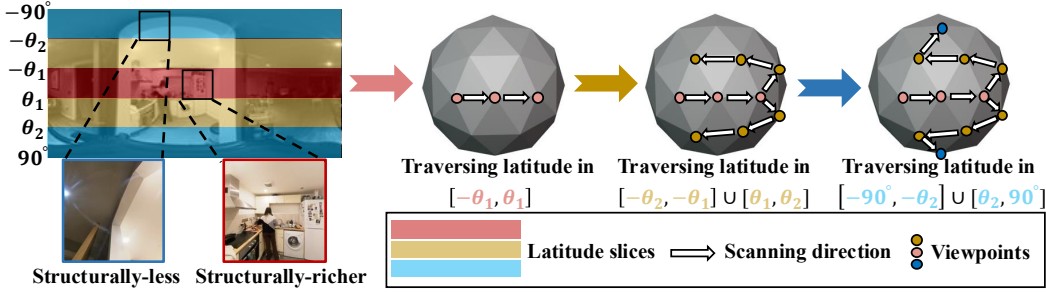

Figure 3: Illustration of the proposed latitude-aware traversing strategy, with K set to 2 as an example.

contain richer structural details; and (2) it captures spherical neighborhood relationships, facilitating the generation of more continuous video sequences, as discussed in Sec. 3.3.

### 3.3 Latitude-Guided Scanning (LGS) Strategy

After obtaining $N$ viewpoints using the SUS strategy, our objective is to organize these viewpoints to obtain video frames $\{\mathbf{F}_1, \ldots, \mathbf{F}_N\}$ that can be effectively compatible with VDE models.

**Effect of starting latitude.** The conventional scanning path of viewpoints starts from the north pole and progresses toward the south pole. Consequently, the initial viewpoints predominantly cover polar regions with less structural details. However, VDE models rely heavily on initial frames to ensure accurate predictions in subsequent frames [22]. Therefore, as depicted in Fig. 4, we investigate how the starting latitude of the scanning path influences performance. As demonstrated at the bottom of Fig. 4, performance improves significantly as the absolute value of the starting latitude decreases. This finding underscores the importance of prioritizing viewpoints from low-latitude regions early in the video sequence. Notably, for a given absolute latitude, initiating the scan in the upper hemisphere (negative latitudes) generally yields superior performance compared to starting in the lower hemisphere (positive latitudes). This phenomenon may be attributed to the earlier placement of information-rich equator regions in the sequence when starting from the upper hemisphere. These insights motivate our subsequent designs.

**Latitude-aware traversing.** To organize information-rich regions early in the video sequence, we propose the latitude-aware traversing strategy to fully utilize the structural information in ERP images. Specifically, we define $K$ key latitudes $0 < \theta_1 < \cdots < \theta_K \leq 90°$. As illustrated in Fig. 3, the scanning proceeds multiple traversals: First, viewpoints whose latitude satisfies $|\theta| \leq \theta_1$ are arranged at the beginning of the video sequence. This ensures that viewpoints near the equator are prioritized and placed early. After traversing the low-latitude slice, we move to higher-latitude slices. Viewpoints whose latitude falls within the range $\theta_1 < |\theta| \leq \theta_2$ are then added to the sequence. The traversal continues progressively, from $\theta_1$ to $\theta_K$, until all viewpoints are incorporated into the sequence. In this way, we provide the VDE model with a sequence of frames that more accurately reflect the geometric structure of a 360° image.

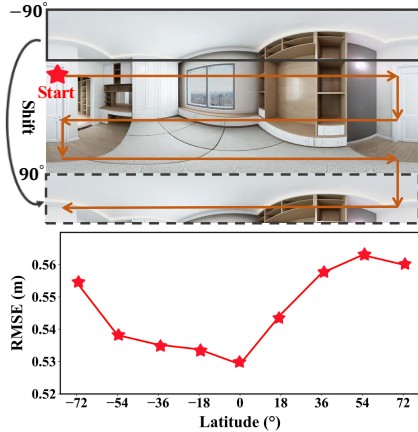

Figure 4: **Top:** Vertically shift the ERP image. **Bottom:** The influence of performance by varying starting latitude.

**Spherical neighbor viewpoint searching.** Based on the SUS strategy, directly searching for the next viewpoint along the horizontal or vertical directions on the ERP plane does not guarantee that the next viewpoint is a spherical neighbor of the current one, as depicted in Fig. 2(b). To ensure a coherent video sequence, we propose searching for the next viewpoint among the spherical neighbors of the current viewpoint. From these neighbors, we select the viewpoint with the lowest absolute latitude, in accordance with the latitude-aware traversing strategy. The spherical neighbor relationships are constructed using ICOSAP, where neighbors are defined as vertices within the same triangular face (See Fig. 2(a)). In practice, the searching process may encounter dead ends. In such cases, we

Table 1: Zero-shot comparison on Matterport3D and Stanford2D3D datasets with $504 \times 1008$ input resolution, following [25]. Numbers are excerpted from [25]. Highlighting: **best**, **second-best**.

| Methods | Matterport3D [15] | | | | | Stanford2D3D [16] | | | | |
|---|---|---|---|---|---|---|---|---|---|---|
| | *AbsRel* ↓ | *RMSE* ↓ | $\delta_1$ ↑ | $\delta_2$ ↑ | $\delta_3$ ↑ | *AbsRel* ↓ | *RMSE* ↓ | $\delta_1$ ↑ | $\delta_2$ ↑ | $\delta_3$ ↑ |
| Marigold [34] | 0.2103 | 0.5745 | 65.46 | 91.36 | 98.19 | 0.2533 | 0.5069 | 58.78 | 87.60 | 96.57 |
| DAv2-Small [33] | 0.2113 | 0.6063 | 65.44 | 91.58 | 98.00 | 0.2343 | 0.5041 | 62.25 | 89.37 | 96.90 |
| PanDA-Small [25] | **0.1206** | **0.4915** | **86.89** | **96.57** | **98.60** | **0.1250** | **0.3462** | **83.60** | **97.05** | **99.46** |
| Ours (VDA-Small) | **0.1408** | **0.4670** | 84.17 | 96.08 | **98.66** | 0.1288 | 0.3529 | **86.76** | 96.60 | 98.89 |
| DAv2-Large [33] | 0.1962 | 0.5522 | 68.37 | 93.03 | 98.28 | 0.2363 | 0.4884 | 60.57 | 88.77 | 96.89 |
| PanDA-Large [25] | **0.1122** | 0.4690 | 88.65 | **97.00** | **98.77** | 0.1026 | 0.3260 | 88.98 | 96.84 | **99.36** |
| Ours (VDA-Large) | **0.1153** | **0.4284** | **89.23** | 96.38 | 98.49 | **0.1005** | **0.2986** | **91.16** | **97.99** | 99.29 |

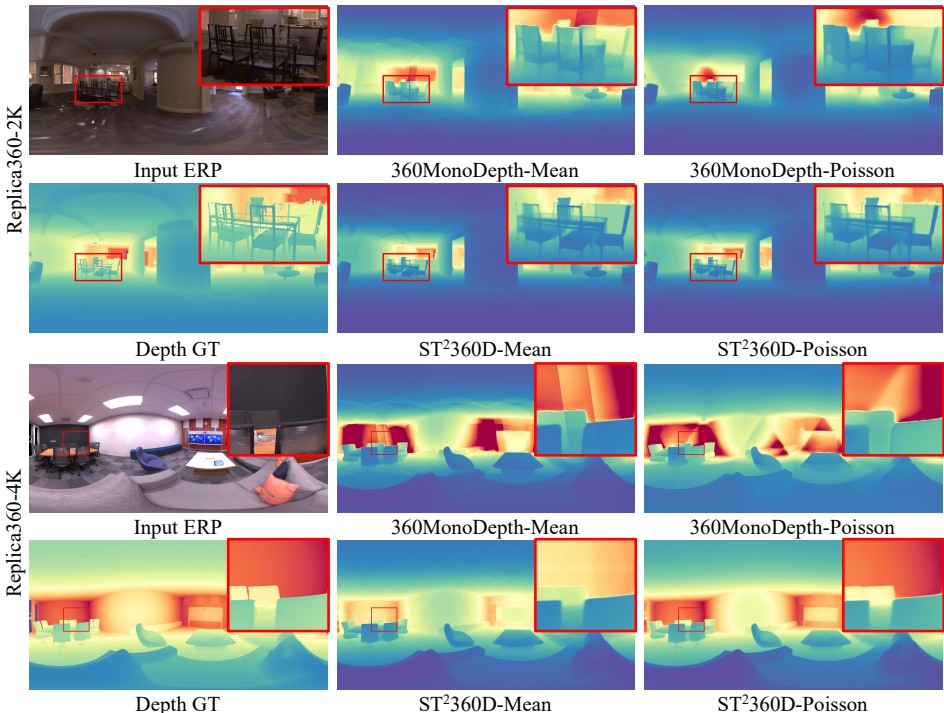

Figure 5: Qualitative results on Replica360-2K (top) and Replica360-4K (bottom) datasets.

manually include all unexplored viewpoints as neighbors of the current viewpoint. This spherical searching ensures the construction of a coherent video sequence.

### 3.4 Video Depth Estimation and Blending

**Video depth map estimation.** We consider VDA [22] as a representative example of VDE models. During the pre-processing phase, VDA divides the input video frames into overlapping segments. The final frame of the last segment is optionally repeated to make it compatible with the temporal dimension. Each segment of video frames is processed by VDA to generate the corresponding disparity maps. In the post-processing stage, adjacent segments are aligned based on a shared scale and shift, determined by two common key frames. The segments are then seamlessly integrated to maintain the original length of the video sequence. In our implementation, the disparity maps are converted into depth maps before being re-projected to the ERP plane. Our empirical findings indicate that merging perspective video predictions in depth space yields better quantitative results.

**Video depth map blending.** After re-projecting video depth maps onto the ERP plane, we use mean blending to efficiently recombine them. For each pixel on the ERP plane, its depth value is calculated

Table 2: Quantitative comparison on high-resolution datasets. [M] is "Mean Blending"; [P] is "Poisson Blending". Numbers are excerpted from [19]. Highlighting: **best**, **second-best**.

| Dataset | Method | RMSE↓ | MAE↓ | AbsRel↓ | RMSE-log↓ | $\delta_1$↑ | $\delta_2$↑ | $\delta_3$↑ |
|---|---|---|---|---|---|---|---|---|
| Matterport3D-2K | HoHoNet [49] | 0.4707 | 0.2620 | **0.0967** | **0.0629** | **90.50** | **97.27** | 97.09 |
| | SliceNet [50] | **0.4463** | **0.2153** | **0.0665** | **0.0513** | **95.17** | **98.07** | **99.54** |
| | UniFuse [12] | 0.6040 | 0.3309 | 0.1110 | 0.0728 | 87.79 | 95.70 | 98.38 |
| | PanDA-Small [25] | 0.4868 | 0.2770 | 0.1311 | 0.0771 | 85.03 | 96.18 | 98.47 |
| | PanDA-Base [25] | 0.4813 | 0.2747 | 0.1288 | 0.0755 | 85.34 | 96.13 | 98.54 |
| | PanDA-Large [25] | 0.4658 | 0.2584 | 0.1190 | 0.0713 | 87.27 | 96.55 | **98.59** |
| | 360MonoDepth [18] | 0.7729 | 0.5106 | 0.2653 | 0.1253 | 60.38 | 85.55 | 94.70 |
| | 360MonoDepth (DAv2) | 0.7968 | 0.4401 | 0.1822 | 0.1076 | 71.99 | 91.86 | 96.78 |
| | Peng *et al.* [19] | 0.4791 | 0.2655 | 0.1004 | 0.0662 | 90.23 | 97.09 | 98.93 |
| | Ours (VDA-Small)[M] | 0.6055 | 0.4033 | 0.2138 | 0.1069 | 71.07 | 90.86 | 96.75 |
| | Ours (VDA-Large)[M] | 0.5110 | 0.3317 | 0.1718 | 0.0907 | 79.60 | 94.10 | 97.59 |
| | Ours (VDA-Small)[P] | 0.4964 | 0.3044 | 0.1624 | 0.0870 | 81.30 | 93.93 | 97.65 |
| | Ours (VDA-Large)[P] | **0.4460** | **0.2568** | 0.1264 | 0.0732 | 87.41 | 95.01 | 97.80 |
| Replica360-2K | HoHoNet [49] | 0.0300 | 0.0193 | 0.1116 | 0.0671 | 90.31 | 95.90 | 98.11 |
| | SliceNet [50] | 0.0403 | 0.0279 | 0.1590 | 0.0896 | 85.15 | 93.88 | 96.44 |
| | UniFuse [12] | 0.0362 | 0.0248 | 0.1336 | 0.0774 | 86.87 | 95.94 | 97.72 |
| | PanDA-Small [25] | 0.0215 | 0.0140 | 0.0751 | 0.0447 | 95.20 | 99.25 | 99.74 |
| | PanDA-Base [25] | 0.0195 | 0.0127 | 0.0692 | 0.0415 | 96.26 | 99.38 | 99.79 |
| | PanDA-Large [25] | 0.0187 | 0.0119 | **0.0648** | 0.0393 | **96.80** | 99.41 | 99.83 |
| | 360MonoDepth [18] | 0.0706 | 0.0456 | 0.1813 | 0.0865 | 78.48 | 93.56 | 98.34 |
| | 360MonoDepth (DAv2) | 0.0497 | 0.0317 | 0.1407 | 0.0763 | 81.68 | 96.59 | 99.35 |
| | Peng *et al.* [19] | 0.0272 | 0.0182 | 0.1074 | 0.0643 | 90.98 | 96.07 | 98.28 |
| | Ours (VDA-Small)[M] | 0.0241 | 0.0181 | 0.1099 | 0.0581 | 90.17 | 98.47 | 99.79 |
| | Ours (VDA-Large)[M] | 0.0201 | 0.0153 | 0.0882 | 0.0472 | 94.41 | 99.55 | **99.96** |
| | Ours (VDA-Small)[P] | **0.0163** | **0.0116** | 0.0669 | **0.0385** | 96.60 | **99.57** | 99.90 |
| | Ours (VDA-Large)[P] | **0.0144** | **0.0097** | **0.0503** | **0.0296** | **99.07** | **99.84** | **99.97** |
| Replica360-4K | HoHoNet [49] | 0.0357 | 0.0249 | 0.1359 | 0.0744 | 85.17 | 94.63 | 96.61 |
| | SliceNet [50] | 0.0473 | 0.0341 | 0.1891 | 0.0994 | 78.31 | 93.17 | 96.77 |
| | UniFuse [12] | 0.0394 | 0.0289 | 0.1480 | 0.0818 | 82.20 | 96.26 | 98.54 |
| | PanDA-Small [25] | 0.0213 | 0.0141 | 0.0739 | 0.0429 | 96.16 | 99.52 | 99.85 |
| | PanDA-Base [25] | 0.0194 | 0.0128 | 0.0676 | 0.0402 | 96.77 | 99.51 | 99.86 |
| | PanDA-Large [25] | 0.0183 | 0.0118 | 0.0612 | 0.0365 | 97.52 | 99.71 | 99.92 |
| | 360MonoDepth [18] | 0.0611 | 0.0400 | 0.1667 | 0.0815 | 80.04 | 95.25 | 98.47 |
| | 360MonoDepth (DAv2) | 0.0448 | 0.0288 | 0.1264 | 0.0689 | 84.98 | 97.54 | 99.69 |
| | Peng *et al.* [19] | 0.0332 | 0.0239 | 0.1309 | 0.0709 | 86.07 | 94.98 | 96.76 |
| | Ours (VDA-Small)[M] | 0.0231 | 0.0167 | 0.0954 | 0.0510 | 92.20 | 99.15 | 99.96 |
| | Ours (VDA-Large)[M] | 0.0190 | 0.0140 | 0.0774 | 0.0422 | 96.61 | 99.81 | 99.96 |
| | Ours (VDA-Small)[P] | **0.0157** | **0.0110** | **0.0602** | **0.0341** | **98.19** | **99.83** | **99.97** |
| | Ours (VDA-Large)[P] | **0.0144** | **0.0096** | **0.0492** | **0.0292** | **99.16** | **99.84** | 99.96 |

by averaging the values from all re-projected video depth maps covering that pixel. Although ST$^2$360D significantly improves spatial consistency in overlapping regions, minor visible seams may still persist in texture-less areas (See Fig. 5). To mitigate this, we further incorporate Poisson blending [48], which utilizes both first-order (gradient) and second-order (Laplacian) derivatives from the re-projected depth maps, ensuring smoothness of the final 360° depth map.

# 4 Experiment

## 4.1 Experiment Setup

**Datasets.** We evaluate on five 360° depth datasets with varying resolutions. We use Matterport3D [15] and Stanford2D3D [16] at $512 \times 1024$ resolution ($504 \times 1008$ in Tab. 1); Matterport3D-2K [15] and Replica360-2K [51] at $1024 \times 2048$; and Replica360-4K [51] at the highest resolution of $2048 \times 4096$.

Table 3: Ablation on **the effectiveness of temporal consistency** in VDE models.

| Methods | Matterport3D | | | Replica360-2K | | | Replica360-4K | | |
|---|---|---|---|---|---|---|---|---|---|
| | $AbsRel\downarrow$ | $RMSE\downarrow$ | $\delta_1\uparrow$ | $AbsRel\downarrow$ | $RMSE\downarrow$ | $\delta_1\uparrow$ | $AbsRel\downarrow$ | $RMSE\downarrow$ | $\delta_1\uparrow$ |
| Per frame input | 0.1634 | 0.5003 | 79.42 | 0.0985 | 0.0208 | 91.42 | 0.0909 | 0.0202 | 92.47 |
| Video frames input | **0.1403** | **0.4664** | **84.31** | **0.0669** | **0.0163** | **96.60** | **0.0602** | **0.0157** | **98.19** |
| $\Delta$ | 14.14% | 6.78% | 4.89% | 32.08% | 21.63% | 5.18% | 33.77% | 22.28% | 5.72% |

**Implementation details.** By default, we use VDA [22] with ViT-Small as the backbone, which is of feed-forward architecture. Since our pipeline is training-free, the parameters of VDE models are kept frozen during inference. All experiments are conducted on a single NVIDIA A40 GPU.

**Evaluation metrics.** Following [18, 12], we utilize standard metrics for depth estimation: Absolute Relative Error (*AbsRel*), Root Mean Squared Error (*RMSE*), Mean Absolute Error (*MAE*), Root Mean Squared Logarithmic Error (RMSE-log), and three threshold percentage metrics $\delta_t$ for $t \in \{1.25^1, 1.25^2, 1.25^3\}$. Similar to [18, 25], we apply scale-and-shift alignment in the depth space.

## 4.2 Qualitative and Quantitative Evaluation

**Matterport3D and Stanford2D3D.** As shown in Tab. 1, ST$^2$360D consistently outperforms the perspective methods DAv2 [33] and Marigold [34], highlighting its strong zero-shot capability on 360° images. Notably, without any 360° depth annotations for training, ST$^2$360D achieves results comparable to PanDA [25], which fine-tunes DAv2 [33] using synthetic 360° depth datasets [17, 52].

**Matterport3D-2K.** In Tab. 2, our ST$^2$360D performs slightly worse than previous data-specific methods [49, 50]. This is mainly due to that these methods are specifically trained on the Matterport3D dataset at a resolution of $512 \times 1024$, allowing them to leverage dataset-specific characteristics. Additionally, Peng *et al.* [19] outperform our ST$^2$360D by employing a pre-trained 360° depth estimation model [49]. However, when compared with 360MonoDepth [18], which requires no 360° depth annotation, our ST$^2$360D achieves superior performance. Furthermore, we have also replaced the perspective depth estimator used in 360MonoDepth with the recent DAv2. In this case, our ST$^2$360D still surpasses 360MonoDepth. We ascribe it to the temporal consistency in VDE models, which reduces depth inconsistency during inference and benefits the overall performance. To compare with the recent method PanDA [25], we first downsample input 360° images to a resolution of $504 \times 1008$. The resulting depth predictions are then upsampled to the original image resolution using bilinear interpolation, following a similar pre-processing strategy of 360MonoDepth [18]. Our ST$^2$360D outperforms PanDA in most metrics.

**Replica360-2K and Replica360-4K.** In Tab. 2, the results demonstrate that our ST$^2$360D, employing ViT-Small as the backbone with mean blending, already outperforms existing methods across most metrics (12 out of 14) on the Replica360-2K and Replica360-4K datasets. These findings underscore the effectiveness of our method for high-resolution 360° depth estimation, particularly highlighting its robust zero-shot capability. Furthermore, as illustrated in Fig. 5, our ST$^2$360D exhibits more precise structural details compared to 360MonoDepth [18], such as the chairs. Moreover, even when employing only mean blending, our ST$^2$360D significantly reduces visible seams in depth predictions. Our ST$^2$360D outperforms PanDA, which is specifically fine-tuned on synthetic 360° depth datasets.

## 4.3 Ablation Studies

By default, we employ VDA with ViT-Small as the backbone, with $252 \times 252$ patch resolution.

**Effectiveness of temporal consistency.** We conduct an ablation study as presented in Tab. 3, examining the influence of temporal consistency. Specifically, we feed each frame individually into the VDE model, degrading it into a single-image depth estimation model. The results on three datasets consistently demonstrate that temporal consistency significantly enhances overall performance.

**Effectiveness of SUS strategy.** In Fig. 6, using similar number of video frames, the SUS strategy obtains better results than ERP-plane uniform sampling. It is because SUS strategy distributes viewpoints evenly on the spherical surface, avoiding oversampling at the polar regions.

Table 4: Ablation on **the scanning directions** on the Matterport3D dataset.

| Methods | AbsRel ↓ | RMSE ↓ |
|---|---|---|
| Horizontal | 0.1776 | 0.5603 |
| Vertical | 0.1945 | 0.6128 |
| Ours | **0.1530** | **0.5216** |

Table 5: Ablation on **the components of LGS strategy** on the Matterport3D dataset.

| Neigh. | Traverse | AbsRel ↓ | RMSE ↓ |
|---|---|---|---|
|  |  | 0.1776 | 0.5603 |
| ✓ |  | 0.1586 | 0.5307 |
|  | ✓ | 0.1580 | 0.5291 |
| ✓ | ✓ | **0.1530** | **0.5216** |

Table 6: Ablation on **the choice of key latitudes in the LGS strategy** on the Matterport3D dataset.

| Choice | AbsRel ↓ | RMSE ↓ |
|---|---|---|
| {18} | 0.1544 | 0.5246 |
| {36} | 0.1542 | 0.5236 |
| {18, 36} | **0.1530** | **0.5216** |

Table 7: Ablation on **the FoV** of perspective patches on the Matterport3D dataset.

| FoV | AbsRel ↓ | RMSE ↓ |
|---|---|---|
| 60 | 0.1687 | 0.5276 |
| 90 | **0.1530** | **0.5216** |
| 120 | 0.1796 | 0.6028 |

Table 8: Ablation on **the spatial resolution** of perspective patches on various datasets. We report the *RMSE* metric.

| Datasets | $252 \times 252$ | $378 \times 378$ | $518 \times 518$ | $756 \times 756$ |
|---|---|---|---|---|
| Matterport3D | 0.5216 | 0.4664 | **0.4661** | 0.4746 |
| Matterport3D-2K | 0.5497 | 0.5044 | **0.4964** | 0.5016 |
| Replica360-2K | 0.0245 | 0.0170 | **0.0163** | 0.0188 |
| Replica360-4K | 0.0240 | 0.0167 | **0.0157** | 0.0175 |

**Scanning directions.** Tab. 4 presents an ablation study on scanning directions. Horizontal scanning arranges viewpoints horizontally, while vertical scanning arranges viewpoints vertically. Results show horizontal scanning performs better due to smoother transitions. Furthermore, our LGS strategy outperforms both horizontal and vertical strategies, showing the importance of prioritizing structurally-rich regions early in the sequence.

**The LGS strategy.** From Tab. 5, incorporating spherical neighbor viewpoint search improves performance by ensuring continuous scanning on the spherical surface. Employing latitude-aware traversing also enhances performance by prioritizing low-latitude regions. Crucially, integrating both components achieves optimal results. Furthermore, as shown in Fig. 6, the LGS strategy consistently improves overall performance on various frame counts. With both the SUS and LGS strategies, a notable performance gain is observed when increasing the number of video frames from 40 to 160.

**Key latitude in LGS.** We check the effectiveness of key latitude settings within the LGS strategy in Tab. 6. Using a larger key latitude slightly enhances performance by prioritizing more low-latitude slices earlier in the sequence. Moreover, incorporating two key latitudes further improves performance by gradually arranging structurally rich perspective patches into the sequence.

**Impact of FoV.** As shown in Tab. 7, the FoV significantly influences the overall performance. Setting the FoV to 90° achieves optimal performance, whereas increasing it to 120° degrades the performance, likely due to redundant structural information from excessively large views.

**Impact of spatial resolution of patches.** Tab. 8 shows that an intermediate patch resolution of $518 \times 518$, aligned with the training resolution of VDA [22], consistently yields the best performance. Increasing patch resolution from $252 \times 252$ to $378 \times 378$ notably improves performance, but further increases yield minimal benefits and even degrade the performance at a resolution of $756 \times 756$.

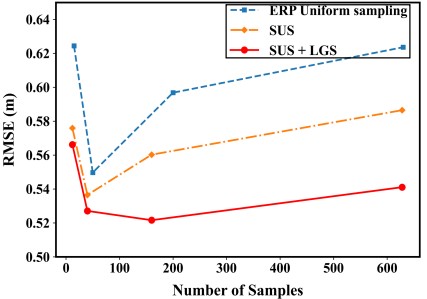

Figure 6: Results using SUS and LGS with varying numbers of frames.

## 4.4 Discussion

**Different VDE models.** Our training-free pipeline is agnostic to specific VDE models. To illustrate this, we evaluate our ST$^2$360D using several VDE models: ChronoDepth [42], DepthCrafter [43], and Depth Any Video [24]. As presented in Tab. 9, our ST$^2$360D consistently outperforms 360MonoDepth [18] with DAv2 as the depth estimator on both Replica360-2K and Replica360-4K datasets.

Table 9: Discussion about employing **different VDE models** in ST$^2$360D. We take 360MonoDepth with DAv2 as the reference and report *RMSE* metric.

| Methods | 2K [51] | 4K [51] |
| --- | --- | --- |
| 360MonoDepth (DAv2) | 0.0497 | 0.0488 |
| ChronoDepth [42] | 0.0278 | 0.0279 |
| DepthCrafter [43] | 0.0260 | 0.0247 |
| Depth Any Video [24] | 0.0399 | 0.0413 |

Table 10: Discussion about **time consumption** (seconds). [M] is "Mean Blending"; [P] is "Poisson Blending".

| Methods | Input Num. | Pre proc. | Inf. | Post proc. | Total |
| --- | --- | --- | --- | --- | --- |
| [18][M] | 20 | 0.6 | 5.4 | 7.8 | 13.8 |
| ST$^2$360D[M] | 12 | 0.3 | 0.7 | 0.1 | 1.1 |
| ST$^2$360D[M] | 40 | 0.7 | 1.0 | 0.1 | 1.8 |
| ST$^2$360D[M] | 160 | 2.4 | 3.0 | 0.5 | 5.9 |
| [18][P] | 20 | 0.6 | 5.4 | 27.7 | 33.7 |
| ST$^2$360D[P] | 160 | 2.4 | 3.0 | 21.5 | 26.9 |

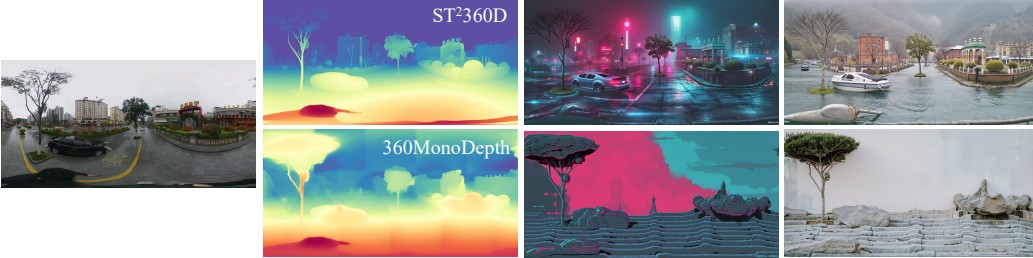

Figure 7: Depth-controlled generation results. First prompt: "cyberpunk style". The second is empty.

**Inference time.** Tab. 10 compares the inference time of our method against 360MonoDepth on the Replica360-2K dataset. Our ST$^2$360D significantly reduces inference time compared to 360MonoDepth. Although adding more video frames increases inference time, our method remains more efficient than 360MonoDepth, which requires additional alignment during post-processing.

**Depth-controlled generation.** In Fig. 7, we showcase depth-controlled generation using 360° depth predictions from our ST$^2$360D and 360MonoDepth. We utilize the ControlNet [53] branch of the FLUX model [54], which takes a depth map to guide the synthesis process. Specifically, we adopt the black-forest-labs/FLUX.1-Depth-dev variant, which supports the use of a referenced depth map as input for conditioning. Due to the stronger zero-shot capability of ST$^2$360D, our generated results consistently surpass those obtained using 360MonoDepth on various prompts.

## 5 Conclusion and Limitation

**Conclusion.** In this work, we propose to represent a 360° image as a sequence of video frames. Accordingly, we introduce ST$^2$360D, a novel training-free pipeline to leverage the inherent temporal consistency of VDE models to enhance spatial consistency across perspective depth patches. To align the perspective frames with VDE models, we further propose two strategies: the spherical-uniform sampling strategy and the latitude-guided scanning strategy. Comprehensive experimental evaluations demonstrate the impressive effectiveness and zero-shot capability of ST$^2$360D in diverse scenarios.

**Broader impacts.** Our ST$^2$360D can provide effective structural priors to support various scene understanding tasks, such as 360° visual navigation, and has the potential to benefit embodied AI.

**Limitation and future work**: Currently, the pipeline of ST$^2$360D is limited to the task of 360° depth estimation. Encouraged by the promising results, future research can extend our ST$^2$360D by incorporating video foundation models in other tasks, such as semantic segmentation.

## Acknowledgments and Disclosure of Funding

This work was supported in part by the National Key R&D Program of China (Grant No.2023YFF0725001), in part by the National Natural Science Foundation of China (Grant No.92370204), in part by the guangdong Basic and Applied Basic Research Foundation (Grant No.2023B1515120057), in part by the Education Bureau of Guangzhou.

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
