# OpenReview forum: "ST$^2$360D: Spatial-to-Temporal Consistency for Training-free 360 Monocular Depth Estimation"
_NeurIPS.cc/2025/Conference — NeurIPS 2025 poster_

### Official Review · Reviewer_zFQ2 · 2025-06-30

**Clarity:** 4
**Significance:** 3
**Originality:** 2
**Rating:** 5
**Confidence:** 4

**Summary:**

ST²360D aims for monocular depth estimation from 360° images by incorporating the video depth estimation model (VDE). The authors first reproject an equirectangular image into perspective patches at sampled regions. Given a sequence of sampled perspective patches, VDE estimates depth with temporal consistency across patches. The estimated patch depths are reprojected to the equirectangular image and mean-/Poisson-blended to get a final 360° depth map. Authors proposed spherical uniform sampling and latitude-guided scanning strategy to reduce redundant sampling at polar region and feed richful information in the initial frame into VDE. These strategies enhance the depth estimation performances of the suggested approach. As a result, ST²360D achieves high zero-shot performance on high resolution images without additional training or fine-tuning.

**Questions:**

1. I suggest including figures that compare depth maps for each comparison method with an error map. This will better depict the geometrical accuracy of ST²360D.
2. I am curious about the depth quality of ST²360D at edge regions. Does temporal consistency and Poisson blending guarantee the sharpness of depth maps then fine-tuning the network with synthetic 360° datasets?
3. In Figure 6, is there an expected reason for degrading accuracy even with SUS and LGS as the number of samples increases? In my sense, temporal consistency across the 360° images should be strengthened thereby reducing the RMSE.

**Ethical Concerns:**

["NO or VERY MINOR ethics concerns only"]

**Final Justification:**

I am satisfied with the clear motivation and the quality the authors have achieved. The authors’ rebuttal and the discussions from other reviewers have further convinced me of the effectiveness of the contribution. I am changing my final score to Accept.

**Limitations:**

Yes

**Quality:**

4

**Strengths And Weaknesses:**

Strength
+ ST²360D investigated the first-time approach of incorporating a video depth estimation model to 360° monocular depth estimation. The authors have suggested simple but effective strategies to enhance 360° depth estimation with existing foundation models. The proposed contributions also improved the depth estimation accuracies.
+ Authors show that temporal consistency by VDE across patches on 360° images is as effective as additional fine-tuning with 360° synthetic datasets (Table 1). Moreover, this method is generalizable since it does not need training or fine-tuning.
+Authors have conducted thorough evaluations and ablation on the method. Experiments show the strength of ST²360D on the zero-shot performance and high-quality results in high-resolution images. Ablation shows the importance of temporal consistency and suggested strategies in the final performance while searching for the best variable sets for achieving the best performance.

Weakness

+ Evaluation is focused on quantitative evaluation, while qualitative comparison is insufficient. Figures mostly show the depth map result of ST²360D variations without visualizing comparisons methods’ depth maps.

---

> ### Author Rebuttal · Authors · 2025-07-31
>
> We sincerely thank you for your positive review and thoughtful questions. "W/Q" numbers the weakness or question, followed by our response.
>
> **W1/Q1: About the qualitative results.**
>
> Thanks for your valuable advice. We will include additional qualitative results for the compared methods, including error maps, to more clearly illustrate the performance differences between our $\text{ST}^2\text{360D}$ and the baselines.
>
> **Q2: About the depth quality at edge regions.**
>
> Thanks for the suggestion. To evaluate the sharpness of estimated depth maps at edge regions, we employ the "*Boundary F1 score*" proposed in Depth Pro [1]. We conduct experiments on the Replica360-2K dataset.
>
> Our method is *training-free* and achieves accurate boundaries by leveraging *temporal consistency and Poisson blending*. We compare our method with PanDA, which *fine-tunes* Depth Anything v2 on *synthetic 360 depth datasets* (Structured3D[2] and Deep360[3]). Note that our method does not support direct fine-tuning as the implementation of Poisson blending is not differentiable.
>
> Given the significant influence of input resolution on boundary accuracy, we compare our method with PanDA at two input resolutions (training resolution of PanDA is 504x1008). Our method takes perspective frames with 518x518 resolution. The results, shown below, demonstrate that our $\text{ST}^2\text{360D}$ with mean blending outperforms PanDA even when using a higher input resolution of 1008×2016. Employing Poisson Blending can further improve the boundary accuracy of our $\text{ST}^2\text{360D}$. We ascribe it as our training-free pipeline is capable of preserving the high depth precision inherent in video depth estimation models, which have been trained on a diverse set of high-quality perspective depth datasets.
>
> | Method        |Input Resolution| Boundary F1 $\uparrow$ |
> |-----------------|:----:|:--------:|
> | PanDA-Small        | 504x1008 | 0.0833  |
> | PanDA-Small        | 1008x2016 | 0.1911  |
> |     Ours (VDA-Small)$^\text{M}$     |518x518 |  0.2134 |
> |     Ours (VDA-Small)$^\text{P}$     |518x518 |  **0.2380** |
>
> **Q3: About the degraded performance with more input views.**
>
> Sorry for the confusion. As shown in Fig. 6, even when applying the SUS and LGS strategies, increasing the number of viewpoints—from 160 to 628—can lead to degraded performance. There are three potential reasons to explain it.
>
> **[1. Limited Visual Transition.]** As the subdivision level increases, the visual difference between adjacent viewpoints becomes minimal. The Video Depth Anything (VDA) model processes 32 frames at a time, and its temporal attention calculation relies on effective motions across frames. When visual transitions are insignificant, it becomes challenging for the model to effectively capture temporal cues.
>
> **[2. Oversampling at Polar Regions.]** Although the SUS strategy has proven effective in reducing oversampling in polar areas, higher subdivision levels inevitably introduce more viewpoints in those regions. This increased sampling density in polar regions may degrade the overall performance.
>
> **[3. Temporal Distance in Long Sequences.]** While the VDA model can handle super-long video sequences, using 628 viewpoints results in a significant temporal gap between the first and last views. This temporal distance may make it challenging to construct coherent spatial relationships between temporally distant viewpoints.
>
> [1] Bochkovskii, Aleksei, et al. "Depth pro: Sharp monocular metric depth in less than a second." ICLR 2025.
>
> [2] Zheng, Jia, et al. "Structured3d: A large photo-realistic dataset for structured 3d modeling." European Conference on Computer Vision. Cham: Springer International Publishing, 2020.
>
> [3] Li, Ming, et al. "MODE: Multi-view omnidirectional depth estimation with 360 cameras." European Conference on Computer Vision. Cham: Springer Nature Switzerland, 2022.

---

> > ### Comment · Reviewer_zFQ2 · 2025-08-06
> >
> > The authors have addressed my questions regarding depth boundary quality and variations in quality depending on input views well. I have no further concerns about this paper and am therefore willing to maintain my positive score.

---

> > > ### Author Response · Authors · 2025-08-06
> > >
> > > Thank you for your comments on our manuscript and response. We sincerely appreciate your positive evaluation.

---

### Official Review · Reviewer_qKmK · 2025-07-02

**Clarity:** 3
**Significance:** 3
**Originality:** 2
**Rating:** 4
**Confidence:** 4

**Summary:**

This paper proposes a novel 360 depth estimation pipeline called ST^2360D. To cover the nature of 360 images that usually contain high-resolution spatial information, the proposed method treats a single 360 image as a sequence of perspective images and successfully utilizes existing video depth estimation methods. The authors suggest two strategies (SUS and LGS) for the seamless adaptation of the video depth estimation model into the 360 image domain. The comprehensive experimental results demonstrate that the proposed pipeline outperforms the previous monocular 360 depth estimation methods and shows the effectiveness of its design choices with two proposed modules.

**Questions:**

- More details about depth-controlled generation experiments are not described in the supplementary material (refer to L310 in the main paper). I wonder how the authors obtained the results in Fig. 7.
- Could you describe the dead-end cases in the viewpoint searching procedure?

**Ethical Concerns:**

["NO or VERY MINOR ethics concerns only"]

**Final Justification:**

The authors have provided thoughtful and appropriate responses to the concerns raised by the reviewers. However, one remaining issue is the novelty of SUS, which I still find unclear. In particular, the sampling strategy based on icosahedral grids does not appear to differ significantly from the method employed in Elite360D.

Despite this, I believe the overall motivation and research direction of the paper are meaningful and have the potential to make a substantial impact. Considering the novelty of the proposed framework and its relevance to the field, I maintain a positive rating for this submission.

**Limitations:**

Yes

**Quality:**

3

**Strengths And Weaknesses:**

### Strengths
- Problem formulation: The authors newly reformulate the 360 depth estimation task as a video depth estimation task and successfully improve the depth quality, especially in terms of depth consistency, where the previous methods suffer from seamless aggregation.

- Training-free pipeline: The proposed pipeline is training-free, and it shows fast inference speed compared to the previous method. Moreover, thanks to its training-free pipeline, the proposed method can easily be expanded to other tasks by utilizing various 4D perception models.

- Well-constructed writing: The paper is well-written and easy to follow. Experimental results are helpful to understand the design choices and effectiveness of each proposed module.


### Weaknesses
- Lack of its technical novelty: Although the proposed pipeline is novel and practical, the technical contribution of the paper is insufficient. Specifically, spherical-uniform sampling (SUS) is already commonly used in 360 3D perception fields (SpherePHD [CVPR’19], PanoFormer [ECCV’22], Elite360D [CVPR’24]).

- Comparison methods: In Table 2, data-specific methods (HoHoNet, SliceNet, UniFuse) are pretty outdated. To validate the effectiveness of the proposed method, it is recommended to compare it with recent 360 depth estimation methods (e.g., HRDFuse [CVPR’23], Elite360D [CVPR’24], PanDA [CVPR’25], etc.).

---

> ### Author Rebuttal · Authors · 2025-07-31
>
> We sincerely thank you for your positive review and thoughtful questions. "W/Q" numbers the weakness or question, followed by our response.
>
> **W1: About the novelty of the SUS module.**
>
> Thank you for your valuable suggestion. We acknowledge the significance of prior work in this domain. Regarding spherical uniform sampling (SUS), please note that it is a sampling mechanism used to represent a sphere. For the previous methods, SpherePHD projects a 360 image onto a spherical polyhedron with triangular faces to minimize spherical distortion; Elite360D maps a 360 image to an icosahedron point set to extract global spherical features; PanoFormer uniformly samples spherical neighbors for pixel-wise self-attention calculation, thereby mitigating distortion.
>
> Sharing a different intuition, the novelty of incorporating SUS in our $\text{ST}^2\text{360D}$ stems from the following:
>
> **1.** The SUS strategy is specifically employed to obtain viewpoints for perspective patch projection, according to the vertices of icosahedral grids at different subdivision levels. This approach reduces redundant sampling in polar regions, which typically contain limited structural information.
>
> **2.** It constructs the spherical relationships among spherical viewpoints, facilitating the spherical neighbor searching in the proposed LGS strategy.
>
> **3.** As demonstrated in Fig. 6 of the main paper, SUS significantly outperforms ERP-plane uniform sampling, showing its effectiveness in aligning 360 images with existing video depth estimation models.
>
> **W2: About the Comparison methods.**
>
> For the mentioned recent 360 depth estimation methods, *HRDFuse* and *Elite360D* have not released their model checkpoints. Therefore, we report the performance of *PanDA* (including its Small, Base, and Large versions) on three high-resolution datasets, as shown in the table below. Following a similar pre-processing strategy of 360MonoDepth, we first downsample input 360 images to a resolution of 504×1008—the training resolution of *PanDA*. The resulting depth predictions are then upsampled to the original image resolution using bilinear interpolation. Our $\text{ST}^2\text{360D}$ outperforms *PanDA* in most metrics.
>
> | Datasets        |Methods| AbsRel $\downarrow$  | RMSE $\downarrow$ |
> |-----------------|----|:--------:|:------------:|
> | Matterport3D-2K        | PanDA-Small | 0.1311  | 0.4868  |
> |          |PanDA-Base |  0.1288 | 0.4813 |
> |          |PanDA-Large | **0.1190** | 0.4658  |
> |          |Ours (VDA-Large)$^\text{P}$ | 0.1264  | **0.4460**  |
> | Replica360-2K        | PanDA-Small | 0.0751  | 0.0215  |
> |          |PanDA-Base | 0.0692  | 0.0195 |
> |          |PanDA-Large | 0.0648 |  0.0187 |
> |          |Ours (VDA-Large)$^\text{P}$ |  **0.0503** | **0.0144**  |
> | Replica360-4K        | PanDA-Small | 0.0739  | 0.0213  |
> |          |PanDA-Base |  0.0676 | 0.0194 |
> |          |PanDA-Large | 0.0612 | 0.0183  |
> |          |Ours (VDA-Large)$^\text{P}$ |  **0.0492** |  **0.0144** |
>
> **Q1: About depth-controlled generation experiments.**
>
> Sorry for the missing explanations. For the depth-controlled generation, we utilize the ControlNet branch of the FLUX model, which takes a depth map to guide the synthesis process. Specifically, we adopt the *black-forest-labs/FLUX.1-Depth-dev* variant, which supports the use of a referenced depth map as input for conditioning. As illustrated in Fig. 7, the generated images preserve consistent spatial relationships across varying text prompts. Specifically, in the third column of Fig. 7, we input the text prompt "cyberpunk style", while in the fourth column, we input the empty text prompt "".
>
> **Q2: About the dead-end cases in the viewpoint searching process.**
>
> We take the case of 160 sampled viewpoints as an example. Each viewpoint is associated with 5 or 6 spherical neighbors. At each stage, the scanning procedure traverses specific horizontal slices of the sphere. For a given viewpoint, if all of its spherical neighbors within the current slice have already been visited, the procedure  may encounter a *dead-end*. In such cases, we expand the neighbor set of the current viewpoint to include all unexplored viewpoints, allowing the scanning procedure to resume. *In Fig. 1 of the supplementary material, we visualize the actual scanning paths with viewpoint indices.* Several dead-end examples can be found in Fig. 1, such as viewpoint transitions 24 $\rightarrow$ 25, 42 $\rightarrow$ 43, and 129 $\rightarrow$ 130. We will clarify the dead-end cases in the revision.

---

> > ### Comment · Reviewer_qKmK · 2025-08-05
> >
> > The authors have provided thoughtful and appropriate responses to the concerns raised by the reviewers. However, one remaining issue is the novelty of SUS, which I still find unclear. In particular, the sampling strategy based on icosahedral grids does not appear to differ significantly from the method employed in Elite360D.
> >
> > Despite this, I believe the overall motivation and research direction of the paper are meaningful and have the potential to make a substantial impact. Considering the novelty of the proposed framework and its relevance to the field, I maintain a positive rating for this submission.

---

> > > ### Author Response · Authors · 2025-08-06
> > >
> > > We sincerely thank you for acknowledging the significance of our research and recognizing the novelty of our proposed framework. We hope our work can offer valuable insights and meaningful contributions to the 360 vision community.
> > >
> > > In response to your concern about the icosahedral sampling strategy, we would like to provide some additional clarifications to make our method clearer.
> > >
> > > **Elite360D** is a dual-branch framework in which the icosahedral point set is taken as the network input that (1) extracts distortion-free fetures to support the ERP branch and (2) adapts naturally to point transformers without relying on hand-crafted spherical convolutions.
> > >
> > > Within our training-free framework, our two proposed strategies **leverage the icosahedral grid to generate a scanning path of viewpoints, transforming a 360 image into a coherent video**. The *SUS* strategy determines *where to sample these viewpoints*, while the *LGS* strategy defines *how to arrange them based on spherical neighborhoods*, *i.e.*, vertices that share the same triangular face in the icosahedral grid.
> > >
> > > We hope this clarification addresses your concern. These points will be modified in our revision.  If you have any further concerns, we would be happy to discuss them. Once again, we sincerely appreciate your positive evaluation.

---

### Official Review · Reviewer_NpNi · 2025-07-02

**Clarity:** 3
**Significance:** 2
**Originality:** 3
**Rating:** 5
**Confidence:** 4

**Summary:**

This paper proposes a training-free method for 360 monocular depth estimation. Existing training-based methods rely on expensive 360 depth estimation and have limited zero-shot capability. Existing training-free methods are time-consuming or rely on 360 depth models. The proposed method addresses these limitations through sequentializing the input equirectangular projection (ERP) image into a sequence of patches and processing the sequence via a pre-trained large zero-shot video depth estimation model. A Spherical-Uniform Sampling (SUS) Strategy and a Latitude-Guided Scanning (LGS) Strategy are proposed to sequentialize the ERP image. Experiment results show that the proposed method possesses comparable zero-shot capability to SOTA training-based methods, and outperforms existing training-free methods by a large margin.

**Questions:**

1. Table 10 shows that 360monodepth takes 20 views as input, which is far less than the proposed method. How will it perform if more views are used?
2. Can you compare the inference time efficiency of your method with more baselines and include GPU memory consumption?

**Ethical Concerns:**

["NO or VERY MINOR ethics concerns only"]

**Final Justification:**

The proposed method has good motivation and performance. The rebuttal shows efficiency comparison results with more baseline methods. And the rebuttal also shows that the proposed method has better scaling property with number of views compared to the baseline method 360monodepth. In all, this is a solid paper and should be accepted.

**Limitations:**

yes

**Quality:**

3

**Strengths And Weaknesses:**

**Strength**
+ The idea of using video depth estimation models to solve the 360 monocular depth estimation task is interesting and novel.
+ The experiment results are impressive, compared to both training-based and training-free methods.
+ Substantial designs of sequentializing strategies are tested.

**Weakness**
+ The illustration of latitude-aware traversing is not clear enough. It is better presented using an algorithm block or in equations.
+ The inference efficiency of the proposed method is only compared with one baseline method. More methods should be included. Furthermore, GPU memory consumption, which is also important, is not given.

---

> ### Author Rebuttal · Authors · 2025-07-31
>
> We sincerely thank you for your positive review and thoughtful questions. "W/Q" numbers the weakness or question, followed by our response.
>
> **W1: About the illustration of latitude-aware traversing.**
>
> Thanks for your suggestion. To make the latitude-aware traversing clear enough, we present it with an algorithm block as follows:
>
> **Algorithm 1: Latitude-aware Traversing**
>
> 1. **Input:** A set of $N$ Viewpoints $\mathbf{v} = \\{\mathbf{v}\_{1},..., \mathbf{v}\_{N}\\}$
>
>   &nbsp;&nbsp;&nbsp;&nbsp;&nbsp;&nbsp;&nbsp;&nbsp;&nbsp;&nbsp; &nbsp;&nbsp;&nbsp;&nbsp;&nbsp;&nbsp;&nbsp; $K$ key latitudes $0 < \theta_1 < \dots < \theta_K \le 90^{\circ}$
>
> 2. **Latitude computation**: For each viewpoint $\mathbf{v}_i$, obtain the  $\Theta(\mathbf{v}_i)$
>
> 3. **Slice partitioning**: Partition viewpoints into latitude slices based on $K$ key latitudes:
>
>     &nbsp;&nbsp;&nbsp;&nbsp;&nbsp;&nbsp;$ \mathbf{v}\_{\text{slice}\_{0}} = \\{\mathbf{v}\_i \mid \mathbf{v}\_i \in \mathbf{v}, |\Theta(\mathbf{v}\_i)| \leq \theta\_{1}\\} $
>
>     &nbsp;&nbsp;&nbsp;&nbsp;&nbsp;&nbsp;$ \mathbf{v}\_{\text{slice}\_{k}} = \\{\mathbf{v}\_i \mid \mathbf{v}\_i \in \mathbf{v}, \theta\_{k} < |\Theta(\mathbf{v}\_i)| \leq \theta\_{k+1}\\} $, for $ k = 1,\ldots,K-1 $
>
>     &nbsp;&nbsp;&nbsp;&nbsp;&nbsp;&nbsp;$ \mathbf{v}\_{\text{slice}\_{K}} = \\{\mathbf{v}\_i \mid \mathbf{v}\_i \in \mathbf{v}, \theta\_{K} < |\Theta(\mathbf{v}\_i)| \leq 90^{\circ}\\} $
> 4. **Slice ordering**: Arrange the slices in ascending latitude to form the list $ \mathbf{v}\_{\text{slices}} \leftarrow [\mathbf{v}\_{\text{slice}\_{0}},\ldots,\mathbf{v}\_{\text{slice}\_{K}}] $
> 5. **Output**: The ordered slices $\mathbf{v}_{\text{slices}}$
>
>
> Based on the latitude slices, we utilize **the spherical neighbor viewpoint searching** to generate the video frame sequence $\\{\mathbf{F}_1, \dots, \mathbf{F}_N\\}$.
>
> **W2/Q2: About the inference time and GPU consumption.**
>
> Thanks for your suggestion. We have extended our evaluation to include inference time and GPU memory consumption across additional baseline methods. All experiments are conducted on a single A40 GPU. The input resolution of UniFuse and ACDNet is 512x1024, while the input resolution of PanDA (Small, Base, and Large variants) is 504x1008. We report the results of our $\text{ST}^2\text{360D}$ with 160 input views and perspective resolution 378x378.
>
> | Method  |Inference Time (s)| GPU memory (G) |
> |-----------------|:----:|:--------:|
> | PanoFormer         | 0.1 |  2.4 |
> | EGFormer         | 0.1 |  2.2 |
> | ACDNet       | 0.07 | 1.7  |
> | PanDA-Small        | 0.06 |  1.2 |
> | PanDA-Base        | 0.1 | 2.3  |
> | PanDA-Large        | 0.3 | 4.4 |
> | 360MonoDepth        | 13.8 | 0.9 |
> | Ours (VDA-Small)     | 5.9 |  1.5 |
>
> **Q1: About more views of 360MonoDepth.**
>
> As suggested, we conduct an extended comparison with 360MonoDepth using increased numbers of input views. In 360MonoDepth, viewpoints are derived from the triangular faces of an icosahedron. A triangular face can be divided into four sub-faces, thus increasing the input views accordingly. *However, the original codebase supports only an unsubdivided icosahedron, and this limitation applies to its input processing, depth estimation, post-processing, and output processing.*
>
> To enable additional input views, we incorporate subdivision functions into the input projection and output re-projection processes by adapting code from SpherePHD [1]. Due to the time constraints of the rebuttal period, we omit the depth alignment module in the post-processing for this evaluation.
>
> We test 360MonoDepth under various subdivision levels: level 0 (20 views), level 1 (80 views), and level 2 (320 views). We utilize the Mean blending, and test on the Replica360-2K dataset. All hyperparameters are set according to the recommended configurations. Our results show that increasing the number of input views degrades the overall performance. *Since each input view is processed independently using an image-based depth estimation model, simply increasing the number of views does not enrich structural information but instead introduces depth inconsistency across views.*
>
> | Input Views     | AbsRel $\downarrow$  | RMSE $\downarrow$ |
> |:--:|:----:|:--------:|
> | 20       | **0.2124** |  **0.0665** |
> | 80        | 0.3805 | 0.0991 |
> | 320     | 0.3699 |  0.1083 |
>
> Moreover, in Fig. 6 of the main paper, we have presented the results of our $\text{ST}^2\text{360D}$ in different input views, ranging from 12 views to 628 views. The results demonstrate the effectiveness of our proposed SUS and LGS strategy, especially when increasing input views from 40 to 160.
>
> [1] Lee, Yeonkun, et al. "Spherephd: Applying cnns on a spherical polyhedron representation of 360deg images." Proceedings of the IEEE/CVF Conference on Computer Vision and Pattern Recognition. 2019.

---

> > ### Comment · Reviewer_NpNi · 2025-08-06
> >
> > Thank you for your responses. All my questions have been addressed. I will raise my rating to 5.

---

> > > ### Author Response · Authors · 2025-08-06
> > >
> > > We sincerely thank you for taking the time to review our manuscript and for your constructive feedback. We are grateful for your decision to raise your rating.

---

### Official Review · Reviewer_YR1d · 2025-07-02

**Clarity:** 3
**Significance:** 3
**Originality:** 3
**Rating:** 5
**Confidence:** 4

**Summary:**

The paper presents a training-free method for depth estimation from 360-degree images by leveraging video depth estimation models on undistorted image patches. The approach employs spherical uniform sampling and latitude-guided scanning to achieve efficient sampling, avoiding oversampling at polar regions and prioritizing the equator to enhance temporal consistency in depth predictions. Resulting in state-of-the-art performance on Stanford2D3D and Replica360 datasets.

**Questions:**

* How robust is the LGS method when applied to outdoor 360-degree datasets, where high-latitude regions may contain significant information?

* Why does the LGS method incorporate randomness rather than a deterministic scanning strategy? What specific measures could be implemented to ensure more consistent results?

* Table 10 indicates that the proposed method has a significantly shorter inference time compared to 360MonoDepth, despite requiring sequential inference. What factors contribute to this efficiency, given that 360MonoDepth does not rely on sequential processing?

**Ethical Concerns:**

["NO or VERY MINOR ethics concerns only"]

**Final Justification:**

My concerns have been sufficiently addressed in the rebuttal; therefore, I maintain my recommendation.

**Limitations:**

yes.

**Paper Formatting Concerns:**

There are no formatting concerns on this paper.

**Quality:**

4

**Strengths And Weaknesses:**

Strengths

* The method is innovative in its ability to directly utilize established foundational models without requiring additional training, enhancing its practical applicability.
* The use of spherical uniform sampling and latitude-guided scanning effectively ensures temporal consistency, with experimental results demonstrating superior performance compared to existing methods.
* The paper provides thorough experimental evidence to support the novelty and significance of the proposed method, strengthening its contributions.
* The paper is well-structured and accessible, making the methodology and results easy to follow for readers.

Weaknesses
* The latitude-guided scanning method assumes that information near the equator (zero latitude) is dominant and serves as an optimal starting point for achieving temporal consistency. This assumption may not hold for outdoor 360-degree datasets, where regions near 90-degree latitudes could contain critical information, potentially limiting the method’s generalizability.
* The latitude-guided scanning approach introduces randomness (noted in line 209, particularly that the algorithm could encounter a "dead end" during scanning), which may lead to instability or inconsistent results.

---

> ### Author Rebuttal · Authors · 2025-07-31
>
> We sincerely thank you for your positive review and thoughtful questions. "W/Q" numbers the weakness or question, followed by our response.
>
> **W1/Q1: About the robustness of the LGS strategy in outdoor datasets.**
>
> Thank you for your valuable suggestions. Based on additional experiments, we confirm that the proposed LGS strategy demonstrates robustness in outdoor scenarios. Below we describe the details of these additional experiments.
>
> **[Outdoor datasets:]** We utilize two datasets containing outdoor scenes. The first one is the Deep360 dataset [1], including outdoor driving scenes generated via the CARLA simulator. Its testing set contains 600 samples. The second dataset is derived from the Matterport3D dataset. We employ SegFormer [2] to detect sky regions in each sample of the Matterport3D dataset and collect samples where the detected sky regions constitute more than 10% of the entire ERP images. Subsequently, we manually select 100 samples by further filtering out wrongly detected samples.
>
> **[Structural Information Across Latitudes in Outdoor Scenes:]** We quantitatively assess the distribution of structural information across latitudes. Specifically, we utilize the Sobel operator to detect edges of 360 ground-truth depth maps, and normalize edge maps to the range [0, 255]. Subsequently, we average all edge maps within each dataset and summarize the edge values within four distinct latitude slices: $[-90^\circ, -45^\circ)$, $[-45^\circ, 0^\circ)$, $[0^\circ, 45^\circ)$
> , and $[45^\circ, 90^\circ]$. The **summarized edge values** are normalized and presented in the following table. The results indicate that equator regions consistently exhibit rich structural information across both outdoor datasets. Furthermore, high-latitude regions can also contain significant structural information, particularly in the Deep360 dataset.
>
> | Dataset       | Samples | $[-90^\circ, -45^\circ)$   | $[-45^\circ, 0^\circ)$   | $[0^\circ, 45^\circ)$   | $[45^\circ, 90^\circ]$   |
> |---------------|:---------:|:------------:|:------------:|:------------:|:------------:|
> | Deep360       | 600     | 38%  | **41%**  | 19%   | 2%    |
> | Matterport3D  | 100     | 13%   | 35%  | **37%**  | 15%   |
>
> **[Ablation studies of LGS strategy in outdoor scenes:]** To evaluate the robustness of the proposed LGS strategy, we conduct ablation studies on the two outdoor datasets. The results are presented in the table below. These results demonstrate that the LGS strategy consistently improves performance in outdoor scenes.
>
> | Dataset        || AbsRel $\downarrow$  | RMSE $\downarrow$ |
> |-----------------|----|:--------:|:------------:|
> | Matterport3D        | w/o LGS | 0.2725  | 0.9498  |
> |          |w/ LGS | **0.2512**  | **0.8977**  |
> | Deep360        | w/o LGS | 2.3860  | 1.5312  |
> |          |w/ LGS | **2.0369**  | **1.3986**  |
>
> **[Conclusion:]** **(1)** From the results of *structural information*, it can be found that equator regions contain rich structural information in most cases. Thus, starting from equator viewpoints is generally suitable. **(2)** Furthermore, our proposed LGS strategy is still effective in outdoor scenes. Notably, the LGS strategy rearranges perspective views rather than discarding high-latitude views entirely. The high-latitude views can also be aligned with the previous equator views. **(3)** Finally, a promising direction for future work involves designing an adaptive LGS strategy that automatically identifies the most salient region as the starting point.
>
> **W2/Q2: About the randomness of the scanning strategy.**
>
> Sorry for the confusion. In the proposed LGS strategy, given **(1)** the viewpoints determined by the SUS strategy, **(2)** the starting viewpoint, and **(3)** key latitudes, the scanning path becomes entirely deterministic. Notably, we have presented the scanning path using viewpoint indices in Fig. 1 of the supplementary material, including different subdivision levels. A viewpoint might encounter the dead-end case, where no unexplored spherical neighbors remain within the current latitude slice. In this case, all remaining unexplored viewpoints in this latitude slice are collected as the neighbor set for the current viewpoint, allowing the scanning process to continue. From this expanded neighbor set, we select the viewpoint with the minimum latitude as the subsequent viewpoint.
>
> Although such dead-end cases could potentially introduce variability, **they consistently arise at specific viewpoints, and the selection of the next viewpoints remains uniform across implementations**. Moreover, we have verified our experimental results multiple times and across diverse GPU environments to confirm reproducibility and robustness of the scanning process.
>
> **Q3: About the efficiency compared to 360MonoDepth.**
>
> The primary reason for the improved efficiency of our method compared to 360MonoDepth lies in **the avoidance of time-consuming depth map alignment in the post-processing.** In 360MonoDepth, depth alignment involves multi-scale operations and incorporates several optimization terms, such as smoothness and scale consistency, which significantly increase time consumption.
>
> In contrast, our pipeline does not require such alignment, as it leverages the temporal consistency inherent in video depth estimation models during inference. Moreover, our SUS and LGS strategies efficiently organize the perspective views into a coherent video sequence.
>
> Furthermore, The Video Depth Anything (VDA) model used in our pipeline can process 32 frames simultaneously within each sequential step. As the number of input views increases from 12 to 160, the number of sequential processing steps increases from 1 to 8, with a 12-frame overlap between adjacent sequences. This design ensures efficiency even with large number of  input views.
>
> [1] Li, Ming, et al. "MODE: Multi-view omnidirectional depth estimation with 360 cameras." European Conference on Computer Vision. Cham: Springer Nature Switzerland, 2022.
>
> [2] Xie, Enze, et al. "SegFormer: Simple and efficient design for semantic segmentation with transformers." Advances in neural information processing systems 34 (2021): 12077-12090.

---

> > ### Comment · Reviewer_YR1d · 2025-08-05
> >
> > I appreciate your clear and precise rebuttal.
> >
> > Your response has thoroughly addressed my concerns regarding the outdoor dataset, randomness, and inference speed.
> > I believe my questions have been sufficiently answered.

---

> > > ### Author Response · Authors · 2025-08-05
> > >
> > > We sincerely appreciate your time in reviewing our manuscript and responses. Thank you for your positive evaluation.

---

### Decision · Program_Chairs · 2025-09-17

**Decision:**

Accept (poster)

**Comment:**

The paper proposes a training-free method that reformulates 360-degree depth estimation as a video depth estimation problem, supported by two sampling and scanning strategies. Reviewers appreciated the strong empirical performance, clear writing, and practical advantages of leveraging foundation video depth models without retraining. Main concerns were about the limited novelty of the sampling strategy, incomplete comparisons with the very latest baselines, and initially insufficient qualitative analysis. The rebuttal convincingly addressed these points with new experiments on outdoor datasets, additional efficiency and boundary-sharpness evaluations, and clarifications on determinism and novelty. Overall, despite minor reservations, the consensus is that the work is solid, impactful, and should be accepted.